# Fe(III)–Chitosan Microbeads for Adsorptive Removal of Cr(VI) and Phosphate Ions

**Swati A. Tandekar [1], Manoj A. Pande [2], Anita Shekhawat [3], Elvis Fosso-Kankeu [4], Sadanand Pandey [5,\*] and Ravin M. Jugade [3,\*]**

1. Department of Chemistry, Tai Golwalkar Mahavidyalaya, Ramtek 441106, India; swatitandekar30@gmail.com
2. School of Chemical Sciences, KCE'S Moolji Jaitha College, Jalgaon 425002, India; pandemanoj82@gmail.com
3. Department of Chemistry, R.T.M. Nagpur University, Nagpur 440033, India; annu.shekh22@gmail.com
4. Department of Mining Engineering, College of Science, Engineering and Technology, Florida Campus, University of South Africa (UNISA), Johannesburg 1710, South Africa; elvisfosso.ef@gmail.com
5. Department of Chemistry, College of Natural Sciences, Yeungnam University, 280 Daehak-Ro, Gyeongsan 38541, Gyeongbuk, Korea
* Correspondence: sadanand.au@gmail.com or spandey@ynu.ac.kr (S.P.); ravinj2001@yahoo.co.in (R.M.J.)

**Abstract:** Fe(III)–chitosan microbeads (Fe–CTB) were prepared using a chemical coprecipitation method. SEM–EDX, FTIR, XRD, TGA, BET, and pH $_{pzc}$ were performed for the characterization of the adsorbent. Various parameters were optimized as pH, adsorption time, adsorbent dose, initial Cr(VI), and PO$_4^{3-}$ ion concentration and the effect of assorted ions for adsorption studies. Fe–CTB microbeads revealed more than 80% detoxification for a 100 mg L$^{-1}$ initial concentration at pH 3 with 60 min stirring of Cr(VI) and PO$_4^{3-}$ ion having adsorption capacities of 34.15 and 32.27 mg g$^{-1}$, respectively. The adsorption process for Cr(VI) and PO$_4^{3-}$ ion followed the monolayer adsorption as they favored the Langmuir isotherm model. Kinetic and thermodynamic studies' emphasis on the adsorption process was spontaneous and exothermic with pseudo-second-order kinetics for both adsorbates. The microbeads were found to be reusable in multiple cycles.

**Keywords:** chitosan; Fe(III)–chitosan composite; chromate; phosphate; adsorption





## 1. Introduction

Pollutants cover both molecular objects, such as fertilizers, pesticides, and dyes, and elemental contaminants, such as metals and nonmetals. The assimilation of these poisonous particles in the food chain ends in an inhibitory or toxic effect on living organisms. Despite stringent laws limiting their careless disposition, these toxic elements may appear in various wastewaters originating from sewage and industry sludge. Heavy metals are nonbiodegradable and are lethal even at quite low concentrations and favor to gather in living organisms, breeding several disorders and diseases [1].

Cr(III) is known as a biological essential nutrient, whereas Cr(VI) is known for its carcinogenic and genotoxic nature in living beings. Products containing chromates and dichromates on consumption cause allergic dermatitis and irritant dermatitis [2,3]. Skin irritation, kidney damage, liver damage, and gastric damage occur after ingestion of water containing Cr(VI) ions. Effluents from numerous industries, such as tannery, paints, electroplating, dyeing, and petroleum, contain Cr(VI) [4]. According to WHO, the acceptable limit of Cr(VI) is 0.05 mg L$^{-1}$ in potable water [5].

Phosphorus is an essential mineral for the development of microorganisms in ecosystems mainly, but an excess of phosphates in water bodies is one of the major factors causing eutrophication [6]. At present, eutrophication has led to severe problems in open freshwater sources around the world [7]. Phosphate ion concentrations above 0.02 mg L$^{-1}$ in water reservoirs result in eutrophication [8].

Therefore, there is a need for a positive remediation for the detoxification of such toxicants. Removal methodologies include ion exchange [9], chemical precipitation [10], mechanical filtration [11], and redox reaction [12]. The use of chemicals and the generation of toxic sludge that requires further treatment make these techniques expensive and impractical for developing countries. To overcome these issues, the adsorption method is more suitable for the removal of these toxicants as it is cost effective. Various biosorbents, including plant residues and natural biopolymers, have been reported to be extremely useful as adsorbents for detoxification of water [13–17].

Various modified chitosan adsorbents are available in fine powdered form, which are tricky to remove from a solution after adsorption. Hence, to deal with these limitations, trials have been conducted through the formation of chitosan beads or as hydrogels form via various routes [18] that make chitosan responsible for the removal of anionic toxicants effectively. For the structural/functional modification of a chitosan matrix on free amine groups on deacetylated units or hydroxyl groups on carbon, $C_3$ and $C_6$ have been used [19]. Chemical modifications involve grafting, cross-linking, and impregnation with inorganic and organic moieties for Cr(VI) ion adsorption [20]. Meanwhile, physical modifications involve phase modifications of membranes, beads, hydrogels, nanofibers, and so on. Physical and chemical modifications combine increase adsorption efficiency, surface area, and selectivity of chitosan for wastewater treatment [21].

Current research discusses a novel composite formed by the impregnation of Fe(III) ion on a chitosan matrix (Fe–CTB) and developed in the form of homogeneous and uniform spherical beads for the efficient removal of chromate and phosphate ions from a solution through batch adsorption studies.

## 2. Materials and Methods

### 2.1. Materials

All chemicals and reagents used for the synthesis and adsorption process were of analytical grade. The chitosan biopolymer (degree of deacetylation: 95%) was purchased from Merck Chemicals, Ahmedabad. The molecular weight of chitosan was found to be 200 kDa through a viscometric technique. Acetic acid was procured from Merck Ltd., India. Ferric chloride (anhydrous), potassium dichromate, potassium dihydrogen phosphate, and ammonia were from Loba Chemie Pvt. Ltd., Mumbai, India. S.D. Fine Chemicals provided other reagents, such as diphenylcarbazide (DPC), sulfuric acid, nitric acid, hydrochloric acid, sodium hydroxide, sodium nitrate. All the metal-complexing agents were of AR grade.

### 2.2. Synthesis of Fe–CTB Microbeads

The procedure of bead formation used in this article was taken from the literature [22] with some modifications. An amount of 1.0 g of chitosan powder was dissolved in a 2% acetic acid solution with constant stirring for 30 min and formed a thick homogeneous jelly liquid. An amount of 0.5 g of ferric chloride powder was added to the above solution with continuous stirring. Spherical bead synthesis was performed by dripping this solution in a 6.25% ammonia solution using a disposable syringe. The beads were filtered and washed with deionized water until a negative test for chloride ions was obtained. The beads were dried at 50 °C for 12 h; the brown colored beads were converted into black color (Figure 1).

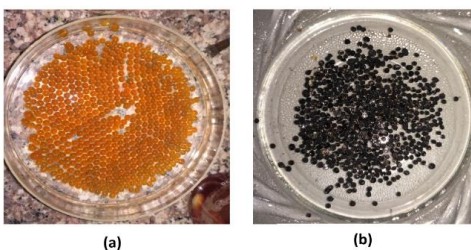

(a) (b)

**Figure 1.** Photographs of Fe–CTB (**a**) before and (**b**) after drying.

### 2.3. Batch Adsorption Experiments

To investigate the performance of Fe–CTB for Cr(VI) and phosphate ion removable from aqueous solutions, batch adsorption experiments were performed by treating different initial Cr(VI) and phosphate ion concentrations ranging from 25 to 300 mg $L^{-1}$ with the adsorbent. The solution's pH was adjusted to 3.0 for both solutions. The solutions were equilibrated with 100 mg of Fe–CTB and stirred with a magnetic stirrer at 298 K for 60 min. The solutions were filtered, and % removal of Cr(VI) and phosphate concentrations were examined in the filtrates. Cr(VI) was estimated spectrophotometrically using a diphenylcarbazide method at 540 nm [23], while phosphate concentration was estimated using the standard ammonium molybdophosphate method at 880 nm [24]. The effects of various working parameters were studied, and the optimum parameters were established independently for Cr(VI) and phosphate ions.

In order to study the effect of pH, the initial pH of the solutions was varied in the range of 2.0 to 7.0 at an initial Cr(VI) and phosphate concentration of 100 mg $L^{-1}$. A known weight of Fe–CTB was added and stirred for 60 min at 298 K. The solutions were filtered, and the solution phase concentration was determined.

To establish the effect of contact time on adsorption efficiency, the experiments were carried out using 100 mg Fe–CTB in 100 mg $L^{-1}$ of Cr(VI) and $PO_4^{3-}$ ion solutions. The stirring time was varied in the range of 5–120 min at 298 K. These solutions were filtered immediately after the specified time intervals and evaluated for the concentrations of the two ions individually.

Adsorbent dose was another parameter that governs the efficiency of adsorption. The adsorbent dosage of Fe–CTB for Cr(VI) and phosphate ion removal from an aqueous medium was examined by varying its quantity from 25 to 500 mg with an initial ion concentration of 100 mg $L^{-1}$ with a stirring time of 60 min and an optimized solution pH 3.0.

The effect of the initial concentration was studied by varying Cr(VI) and phosphate ion concentrations ranging from 25 to 300 mg $L^{-1}$ by keeping all other parameters constant. The final concentration of the ions remaining in the solutions was estimated using spectrophotometry

The amount of Cr(VI) and $PO_4^{3-}$ adsorbed (mg $g^{-1}$) on the beads can be given by the following equation:

$$q_e = \frac{C_o - C_e}{W} \times V \tag{1}$$

where $C_o$ and $C_e$ refer to the initial and equilibrated concentrations in ppm of the adsorbate, $V$ is the volume of the solution in liter, and $W$ is the weight of Fe–CTB beads for adsorption in g. Experiments were performed thrice to check the reproducibility, and the average values were reported.

### 2.4. Characterization

The FTIR analysis of beads was carried out using a Bruker Alpha, London, UK, spectrometer in the range of 500–4000 $cm^{-1}$. SEM analysis was carried out using the Tescan Vega 3 SBH model. X-ray diffractograms were recorded using a Rigaku MiniFlex 300, Tokyo, Japan instrument, while TGA–DTA studies were conducted using a DTG-60 (Shimadzu, Kyoto, Japan) thermal analyzer in nitrogen atmosphere at a 100 mL $min^{-1}$ flow rate. BET analysis was performed using the Quantachrome NOVA 2200e model, which was based on nitrogen adsorption–desorption curves to give surface area and pore properties.

## 3. Results and Discussion

### 3.1. FTIR Spectral Analysis

The native chitosan (Figure 2a) shows distinctive peaks at 3846 and 3266 $cm^{-1}$ of –OH and –NH$_2$ stretching, respectively. The –CH stretching peak was observed at 2890 $cm^{-1}$. The peaks due to N–H and O–H bending were obtained at 1646 and 1423 $cm^{-1}$, respectively. The band at 1151 $cm^{-1}$ was assigned to the pyranose group of chitosan, while the wide band at 1024 $cm^{-1}$ was due to the C–O stretching [25].

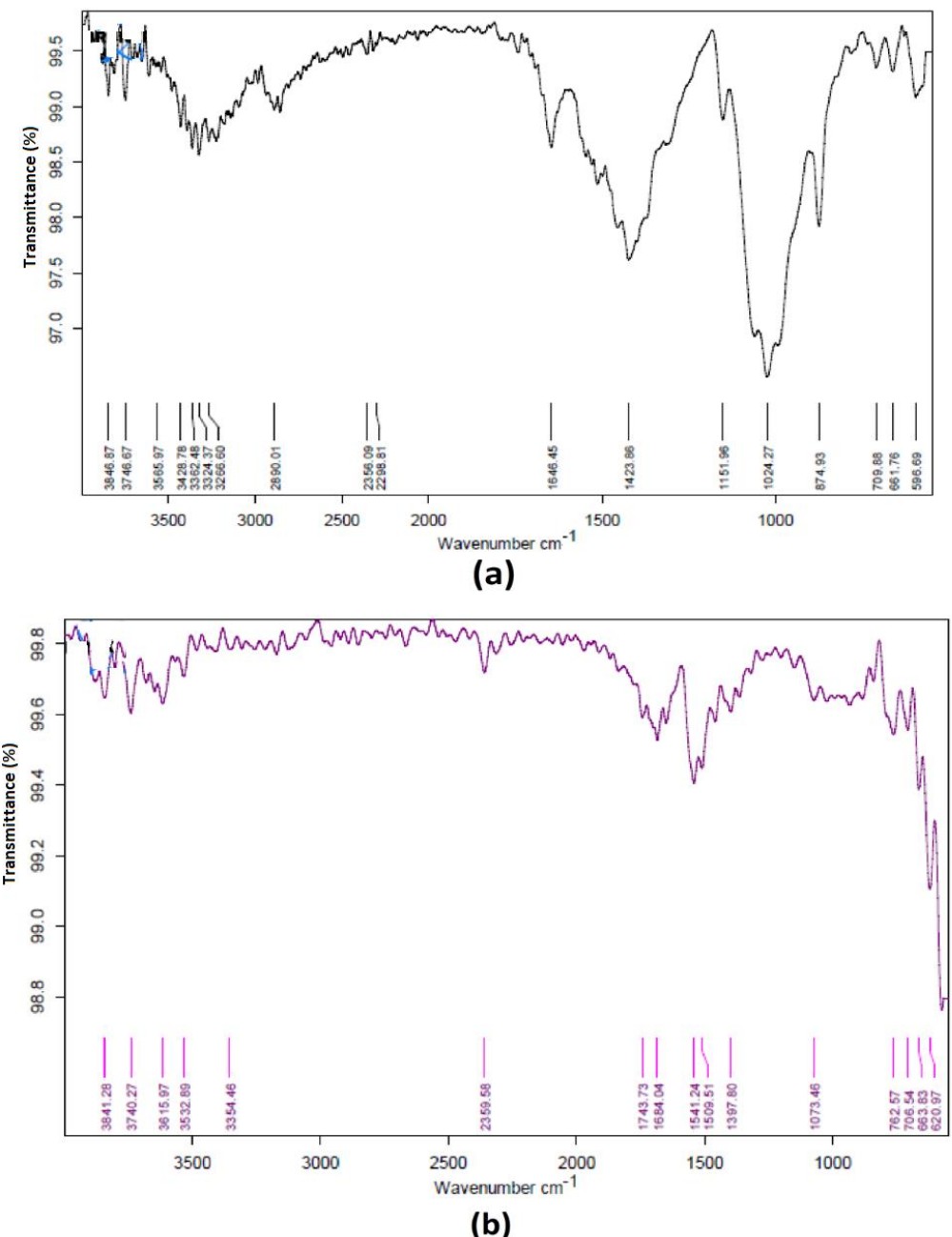

**Figure 2.** FTIR spectra of (**a**) parent chitosan and (**b**) Fe–CTB.

The metal–oxygen (Fe–O) bond in Fe–CTB (Figure 2b) was observed at 620.97 cm$^{-1}$ along with a decreased intensity of other peaks, referring to the functional groups involved in bonding with Fe(III), and hence, the formation of Fe–CTB was confirmed [26,27].

### 3.2. Surface Morphology and Elemental Composition

SEM images taken under different resolutions clearly show morphological and structural changes in Fe–CTB compared with that of native chitosan. The bead size varies from 0.8 to 1.0 mm. The surface morphology shows scaffolds with a heterogeneous structure (Figure 3a). The EDX spectrum showing an Fe peak confirms doping of Fe(III) into the chitosan matrix. Scanning electron micrographs of Fe–CTB with adsorbed Cr(VI) and $PO_4^{3-}$ ion adsorption were recorded, and they depicted subsequent variations in surface morphology. At high resolutions, a rough surface with an ordered folding pattern of Fe–CTB seems to become a smooth and covered surface after Cr(VI) ion adsorption (Figure 3b).

This can be attributed to the anionic charges of chromate ions that destroy the cross-linking in random positions. Similarly, after phosphate ion adsorption, the surface of Fe–CTB was seen to be covered and completely destroyed the folding pattern possibly due to the anionic interaction of $PO_4^{3-}$ ions that cross-linked in various areas of the composite (Figure 3c). EDX studies revealed additional peaks of chromium and phosphorus after adsorption of their ions.

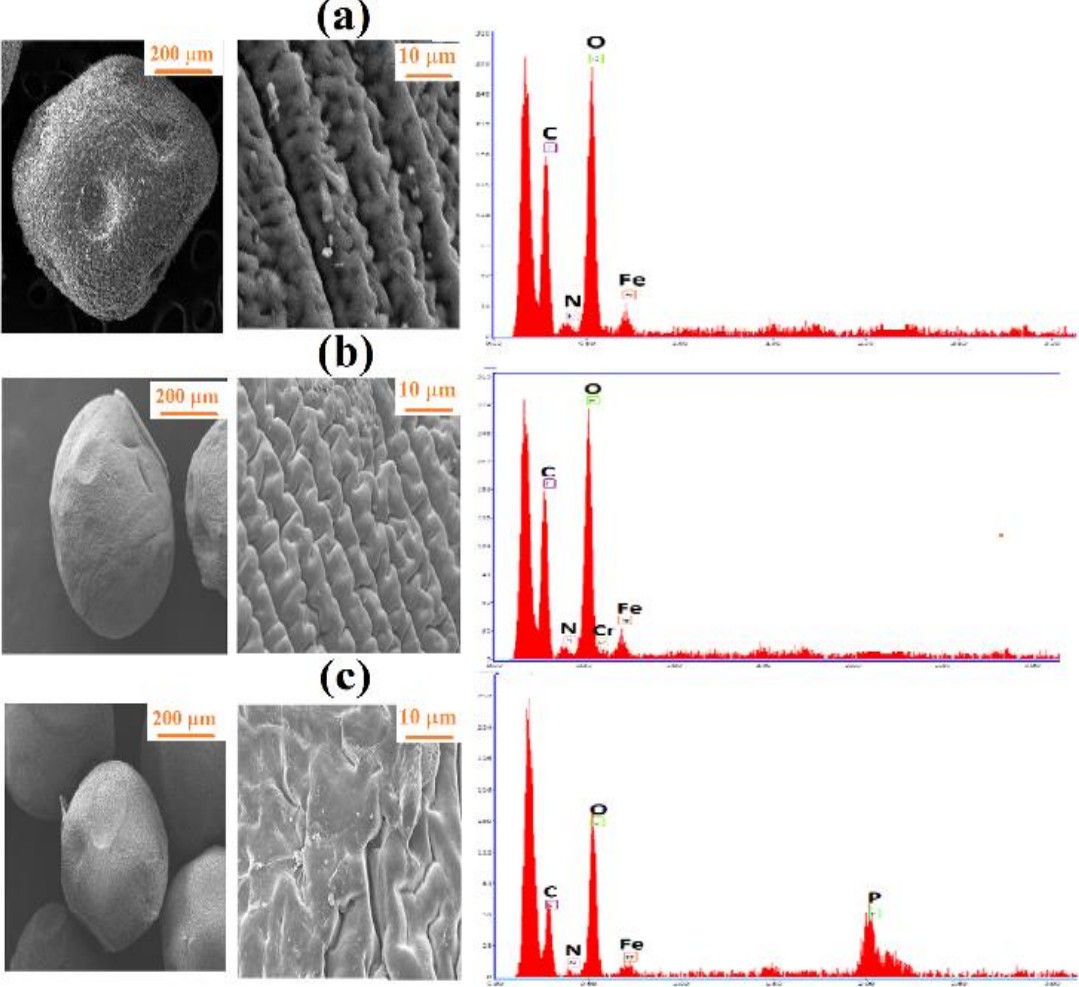

**Figure 3.** (**a**) SEM images at different resolutions and EDX spectra of (**a**) Fe–CTB microbeads (**b**) after Cr(VI) adsorption and (**c**) after phosphate adsorption.

### 3.3. Thermal Studies

TGA–DTA studies were performed to understand the thermal stability of modified chitosan beads. The thermogram of native chitosan (Figure 4a) showed about 10% weight loss at around 100 °C due to dehydration, and the same was observed in the DTA with an endotherm. The second weight loss observed at 300 °C refers to a complete degradation of chitosan due to the breaking of polymeric chain links, and the corresponding exotherm was observed in DTA also. In the case of Fe–CTB, the extent of dehydration was reduced remarkably (Figure 4b). This is due to the fact that the beads were already dried in the hot-air oven. Additionally, the endotherm in the DTA curve was flattened. This Fe–CTB started degrading at around 200 °C with a small exothermic peak at 275 °C. The entirely different TGA curve clearly indicates the formation of modified chitosan beads, which have different thermal properties as compared with native chitosan.

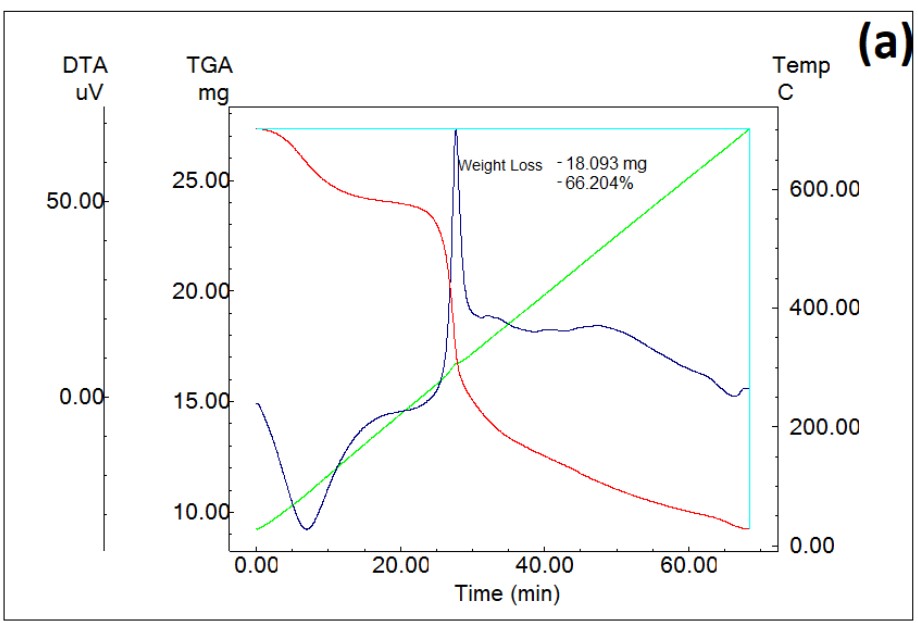

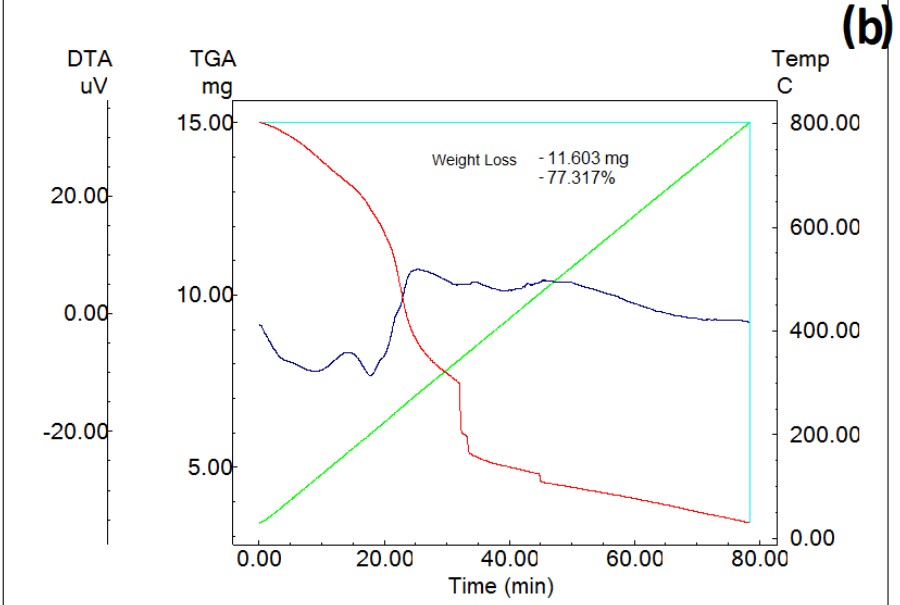

**Figure 4.** TGA and DTA curves of (**a**) parent chitosan and (**b**) Fe–CTB.

### 3.4. BET Surface Area

The surface area of a sorbent is a prime factor because the adsorption process is surface dependent. The surface areas of raw chitosan and Fe–CTB were examined by BET surface area analysis method. The BET surface area of native chitosan was found to be 0.20 m$^2$ g$^{-1}$, whereas it reduced to 0.15 m$^2$ g$^{-1}$ in Fe–CTB. This decrease can be attributed to the fact that in Fe–CTB, the Fe(III) ions crosslink the chitosan, thereby occupying the interstitial sites and also the large particle size of the beads as compared with powdered.

### 3.5. pH Point of Zero Charge (pH$_{PZC}$)

The pH point of zero charge (pH$_{PZC}$) of Fe–CTB was found to be 5.7 by batch equilibrium technique [28]. Amounts of 50 mL of 0.1 M NaCl solutions with 100 mg of Fe–CTB were taken in conical flasks with a varying pH at 2.0–9.0, stirred for 24 h. The solutions were filtered, and the final pH of each system was measured. A graph was plotted between

$\Delta pH$ ($pH_{initial}$-$pH_{final}$) and $pH_{initial}$, whose x-intercept gives $pH_{PZC}$ (Figure 5), indicating that the surface charge of Fe–CTB is positive below pH 5.71 and negative above it.

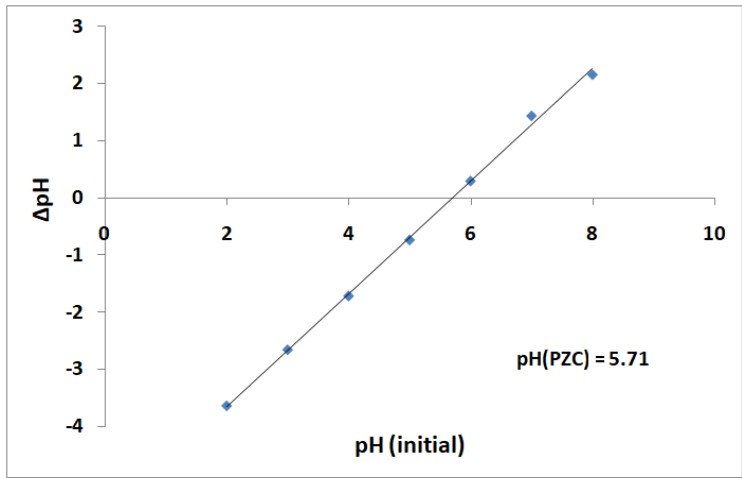

**Figure 5.** pHpzc of Fe–CTB.

*3.6. Batch Adsorption Studies*

3.6.1. Effect of Solution pH

After the adsorption of Cr(VI) and phosphate ions on Fe–CTB under different pH conditions, the adsorption efficiency was maximum at pH $3.0 \pm 0.1$ for both adsorbates (Figure 6a). Hence, pH 3.0 was set and selected for further studies. Additionally, at this pH, Cr(VI) exists as a dichromate ion, which interacts with Fe–CTB, while phosphate ions exist in $HPO_4^{2-}$ form.

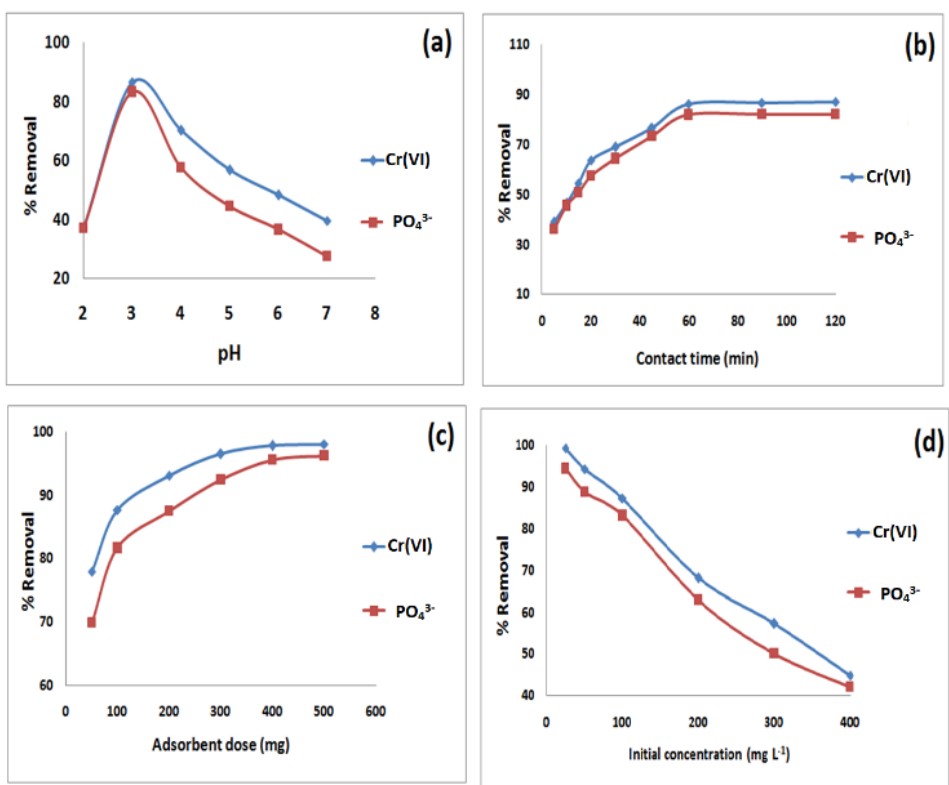

**Figure 6.** (**a**) Effect of initial solution pH, (**b**) effect of contact time, (**c**) effect of adsorbent dose, and (**d**) effect of initial concentration on the adsorption efficiency of Fe–CTB towards Cr(VI) and $PO_4^{3-}$.

### 3.6.2. Effect of Contact Time

The adsorption capacity of Fe–CTB as a function of contact time was investigated in order to attain equilibrium time for maximum uptake and also to study the kinetics of the adsorption process. It was observed that in the initial stages of both systems, the % removal increased with contact time, and the system reached an equilibrium within 60 min. The % removal was observed to be rapid initially and reached equilibrium at nearly 60 min with 86.3% Cr(VI) and 82.1% $PO_4^{3-}$ removal. Hence, 60 min was used for all the subsequent batch studies for both systems (Figure 6b).

### 3.6.3. Effect of Amount of Adsorbent

The results obtained for the variation of adsorbent dose revealed that, for a dosage of 100 mg of Fe–CTB, the adsorption efficiency was more than 80%. A further increase in adsorbent dose led to a very small increase in its efficiency (Figure 6c). Therefore, adsorbent dosage was fixed to 100 mg during all subsequent experiments for both studies.

### 3.6.4. Effect of Initial Concentration

It is shown in Figure 6d that with the increase in sorbates' ion concentration from 25 to 400 mg $L^{-1}$, the adsorption efficiency decreased gradually. This is due to the saturation of the binding sites of Fe–CTB. From Figure 6d, it was observed that around 87.4% and 83.2% removal efficiencies were achieved for 100 mg $L^{-1}$ Cr(VI) and phosphate ion concentration (Figure 6d). Hence, this was used as a fixed concentration for further studies. Higher concentrations led to a much lower adsorption efficiency.

### 3.7. Adsorption Isotherms

An adsorption isotherm study is an appropriate tool to evaluate adsorbents' performance quantitatively. Additionally, it gives information about the adsorption ability and surficial properties and affinity of a prepared adsorbent [29]. The Langmuir [30] and Freundlich [31] adsorption isotherms were applied to study the adsorption nature of Cr(VI) and phosphate ion individually on an Fe–CTB composite (Figure 7). The linearized equations for these two isotherms are given by:

Langmuir isotherm:

$$\frac{C_e}{q_e} = \frac{1}{q_m b} + \frac{C_e}{q_m}$$

Freundlich isotherm:

$$\log q_e = \log K_F + \frac{1}{n} \log C_e$$

These investigations were performed by varying the initial ion concentration (25–300 mg $L^{-1}$) at pH 3.0 with 100 mg Fe–CTB at 298 K and 60 min of stirring time. The data obtained are shown in Table 1. Maximum monolayer adsorption capacities ($q_o$) of 34.15 mg $g^{-1}$ for Cr(VI) and 32.27 mg $g^{-1}$ for phosphate ion were obtained from the Langmuir isotherm model. The $R_L$ values were found to be 0.69 and 0.8, respectively, which are less than 1, indicating a favorable adsorption of sorbate ions on an Fe–CTB surface. The Langmuir isotherm constants ($b$) were found to be 0.0045 and 0.0025, respectively, which correlate with the adsorption potency. The plot of log $q_e$ vs. log $C_e$ gives the constants $k_F$ and $n$ for the Freundlich isotherm model. This interaction allowed for determining the values of $n$, which were 4.138 and 2.99 for Cr(VI) and $PO_4^{3-}$, showing a feasible adsorption performance [32]. The value of the coefficient of determination ($R^2$) implies that adsorption data are properly explained by Langmuir adsorption isotherm models by both systems and have good agreement with experimental $q_e$ with both coefficients of determination close to 1.0. Therefore, this model is best fitted for a study explaining its monolayer adsorption nature on a homogeneous adsorbent surface (Table 1).

## Freundlich Adsorption Isotherm

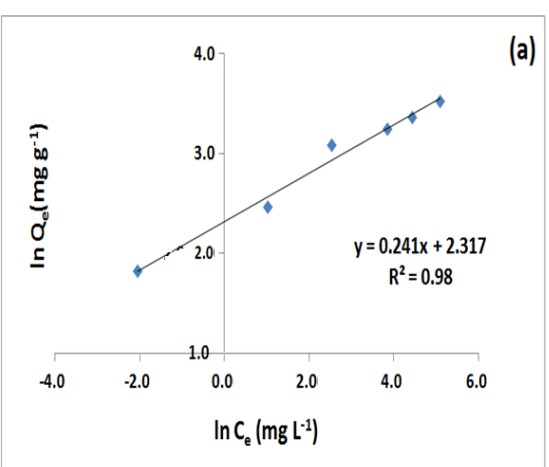
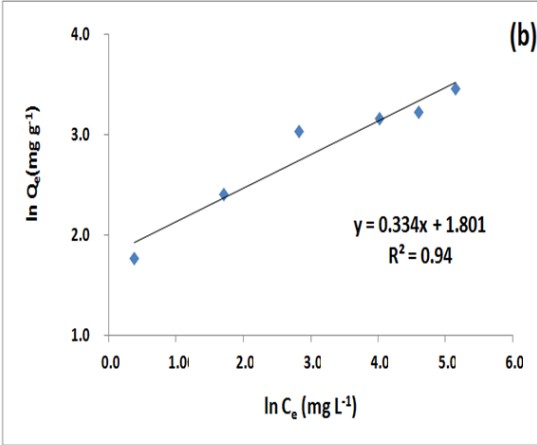

## Langmuir Adsorption Isotherm

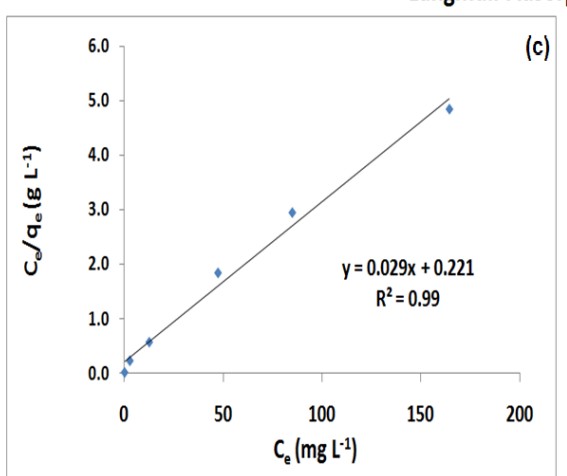
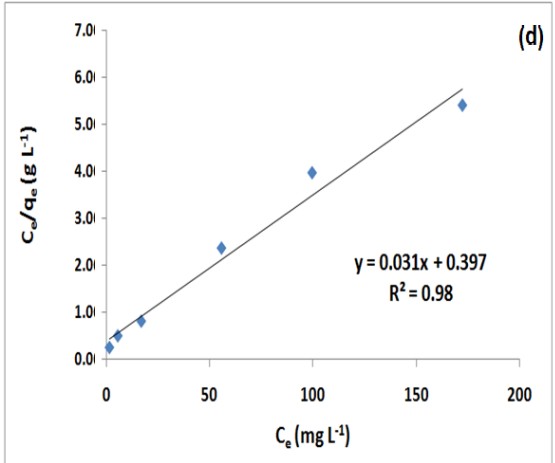

**Figure 7.** Freundlich and Langmuir isotherm models for (**a**,**c**) Cr(VI) and (**b**,**d**) $PO_4^{3-}$.

**Table 1.** Adsorption isotherm model parameters.

| Isotherm | Parameters | Cr(VI) Ion | $PO_4^{3-}$ Ion |
|---|---|---|---|
| Freundlich | kf ($mg^{1-1/n}L^{1/n}/g$) | 10.15 | 6.06 |
|  | n | 4.138 | 2.99 |
|  | R2 | 0.981 | 0.941 |
| Langmuir | qo ($mg\,g^{-1}$) | 34.15 | 32.27 |
|  | b ($L\,mg^{-1}$) | 0.0045 | 0.0025 |
|  | RL | 0.69 | 0.80 |
|  | R2 | 0.992 | 0.984 |

### 3.8. Adsorption Kinetics

Pseudo-first-order and pseudo-second-order models were studied to understand the adsorption process kinetics [33,34]. The best-fit kinetic model was elected based on both the linear coefficient of determination ($R^2$) and the calculated $q_e$ values.

$$\log(q_e - q_t) = \log q_e - \left(\frac{K_1}{2.303}\right)$$

$$\frac{t}{q_t} = \frac{1}{K_2\, q_e^2} + \left(\frac{1}{Q_e}\right)t$$

Graphs obtained between ln $(Q_e - Q_t)$ against time $t$ and $t/Q_t$ against time $t$, according to the pseudo-first-order and pseudo-second-order kinetic model, are shown in Figure 8. It was observed that the adsorption process was rapid initially in 15 min, followed by a sluggish rate until equilibrium was established in about 60 min. It is evident from Table 2 that the pseudo-second-order model was well fitted with the experimental results in terms of $R^2$ as compared with the pseudo-first-order kinetic model for both Cr(VI) and phosphate ion adsorptions individually.

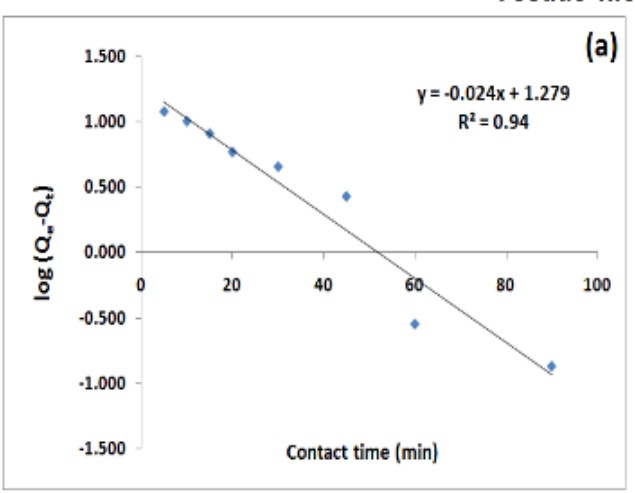
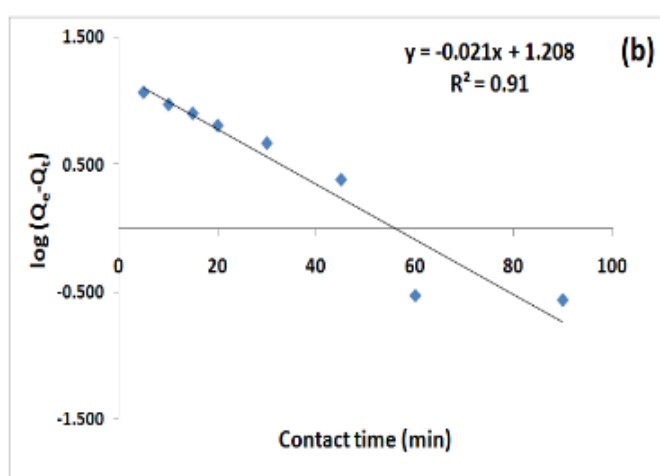
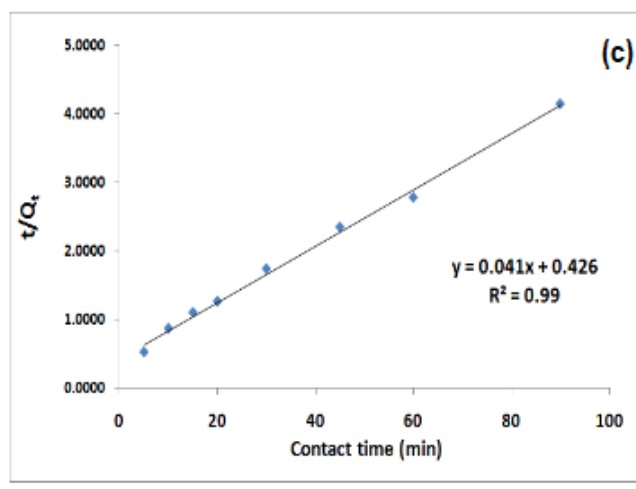
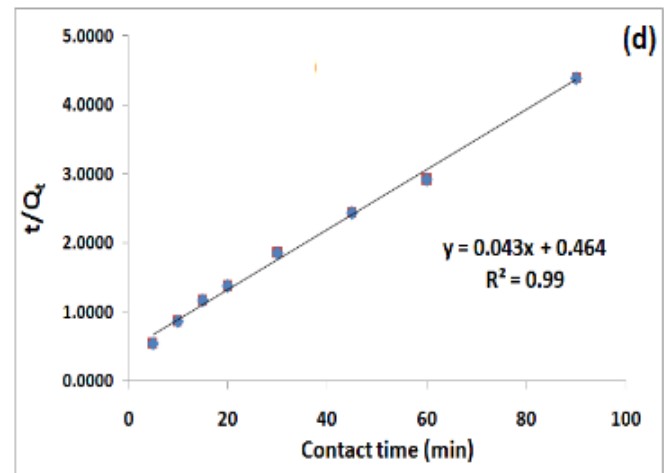

**Figure 8.** Kinetics models for adsorption of (**a**,**c**) Cr(VI) and (**b**,**d**) $PO_4^{3-}$.

**Table 2.** Kinetics of adsorption.

| Rate Model | Parameters | Cr(VI) Ion | $PO_4^{3-}$ Ion |
|---|---|---|---|
| Pseudo-first-order | $K_1$ (min$^{-1}$) | 0.0568 | 0.0497 |
| | $R^2$ | 0.944 | 0.915 |
| Pseudo-second-order | $K_2$ (L mol$^{-1}$ min$^{-1}$) | 0.0040 | 0.0040 |
| | $R^2$ | 0.996 | 0.995 |

### 3.9. Adsorption Thermodynamics

Thermodynamic parameters are useful in establishing the energetics involved in the adsorption process. Adsorption efficiencies were established at different temperatures

varying from 298 to 333 K. Gibbs free energy change was obtained from the equilibrium constant, while enthalpy and entropy changes were obtained using the intercept and slope of the van't Hoff curve (Figure 9). These values are depicted in Table 3. Data of both Cr(VI) & $PO_4^{3-}$ ions indicate negative Gibbs free energy change ($\Delta G$), depicting the spontaneous adsorption behavior that becomes less favorable at a higher temperature. Negative enthalpy change ($\Delta H$) shows an exothermic nature, while negative entropy change ($\Delta S$) clearly indicates a decrease in disorder during the Cr(VI) and phosphate adsorption onto the Fe–CTB surface.

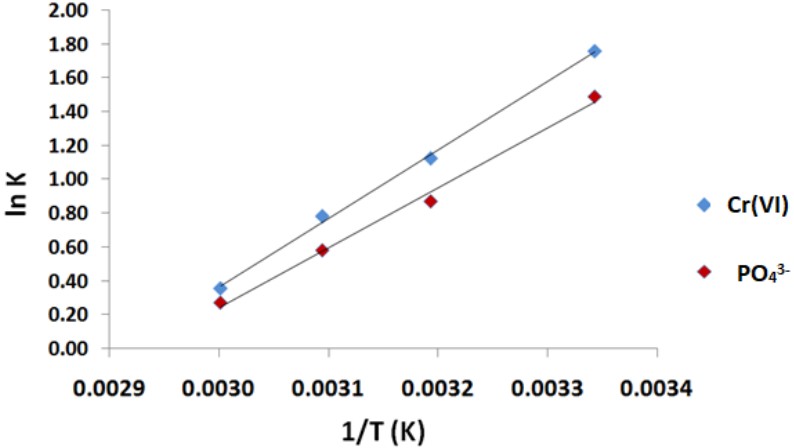

**Figure 9.** van't Hoff plots.

**Table 3.** Thermodynamic parameters.

| Temp. (K) | Cr(VI) | | | $PO_4^{3-}$ | | |
|---|---|---|---|---|---|---|
| | $\Delta G^0$ (kJ mol$^{-1}$) | $\Delta H^0$ (kJ mol$^{-1}$) | $\Delta S^0$ (kJ mol$^{-1}$ K$^{-1}$) | $\Delta G^0$ (kJ mol$^{-1}$) | $\Delta H^0$ (kJ mol$^{-1}$) | $\Delta S^0$ (kJ mol$^{-1}$ K$^{-1}$) |
| 298 | −4.37 | | | −3.57 | | |
| 313 | −3.00 | −33.69 | −0.098 | −2.36 | −29.6 | −0.087 |
| 323 | −2.02 | | | −1.49 | | |
| 333 | −1.04 | | | −0.62 | | |

### 3.10. Diverse Ions Effect

The presence of other co-ions, such as chloride, nitrate, carbonate, and sulfate, which are typically present in targeted contaminated wastewater, gives a negative influence and competes with chromate and phosphate ions for available adsorption sites, hence decreasing the removal efficiency of the prepared Fe–CTB [35]. Thus, it is usual to observe the interference of these diverse ions on removal efficiency in binary solutions. A 100 mg L$^{-1}$ chromium and phosphate solution was mixed with a 100 mg L$^{-1}$ solution of the interfering ion, and the effect of adsorption efficiency was evaluated. It was observed that the existence of co-ions disturbs Cr(VI) and phosphate ion removal capacity (Figure 10a). The competition among the coexisting anions to grab the active surface sites of the adsorbent depends on the concentration, dimension, and charge on the anions [36]. In the present study, $CO_3^{2-}$ had the largest effect on Cr(VI) and phosphate ion adsorption in separate systems, followed by nitrate, phosphate, chloride, and sulfate ions.

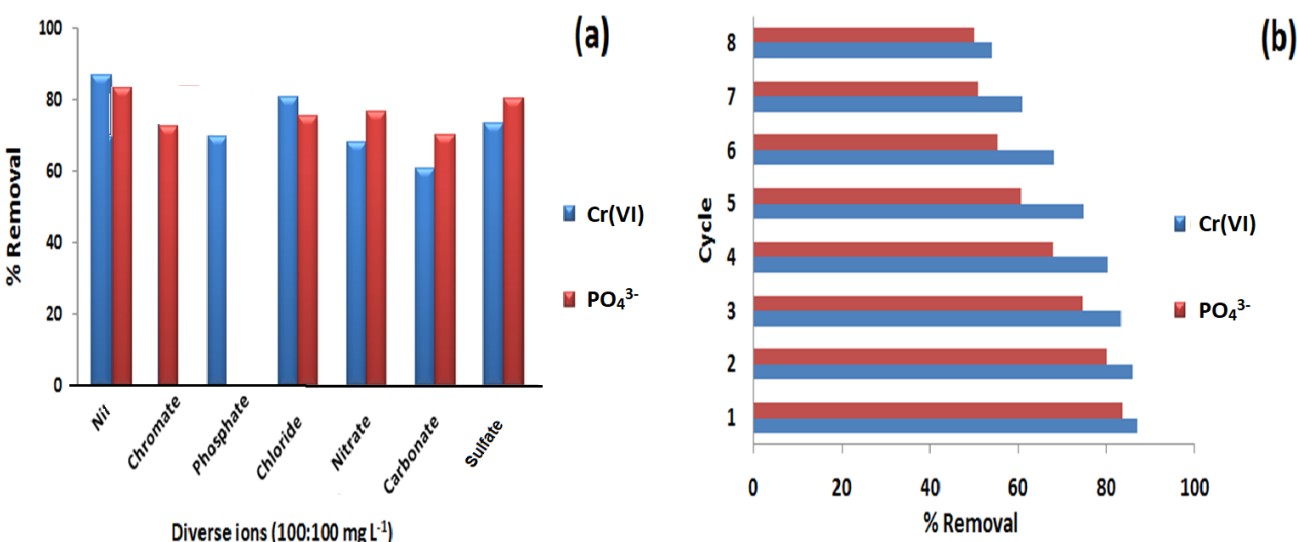

**Figure 10.** (**a**) Effect of diverse ions and (**b**) regeneration cycles after adsorption of Cr(VI) and phosphate ions on Fe–CTB.

### 3.11. Reusability Studies

High adsorption efficiency and good reusability are of prime importance for any type of adsorbent, which would support the economic value of the adsorption method. In the current work, solutions of NaCl, $Na_2SO_4$, $Na_2CO_3$, and $NaNO_3$ were examined for a possible regeneration of the material. It was observed that these reagents effectively regenerate the Fe–CTB complex by ion-exchange reaction. It was evident that the best results were obtained with a 5% *w/v* NaCl solution as a regenerating agent for both toxicants (Figure 10b). Recycling the adsorbent led to a reduction in the adsorption capacities in every successive cycle. The washings obtained after the eighth desorption cycle were subjected to qualitative detection of chromate and phosphate ions. It was observed that the washings contained these ions, indicating that there was an ion-exchange phenomenon taking place, leading to the exchange of chromate or phosphate ions with the chloride ions.

### 3.12. Comparison of Fe–CTB with Reported Materials

Fe–CTB has been compared with various materials in terms of adsorption capacities reported in the recent literature. Table 4 is self-explanatory and gives a clear idea of the impact of the modification of chitosan on the adsorption capacity towards Cr(VI) and $PO_4^{3-}$ ions. It is interesting to note that the native chitosan had adsorption capacities of just 2.48 and 6.65 mg $g^{-1}$ for Cr(VI) and $PO_4^{3-}$, respectively. However, these values were enhanced to 34.15 and 32.27 mg $g^{-1}$, respectively, due to Fe(III) crosslinking with chitosan polymeric chains in Fe–CTB. This proves that the utility of this modification for toxicant sorption was higher and better than all of the materials reported in the literature. Additionally, there are only a few materials reported for the removal of both toxicants. Fe–CTB has the ability to be used for both toxicants.

**Table 4.** Comparison of Fe–CTB for Cr(VI) and phosphate sorption with reported materials.

| Adsorbent | pH | Temp | Time | Adsorption Capacity (mg $g^{-1}$) for | | Reference |
| --- | --- | --- | --- | --- | --- | --- |
| | | | | Cr(VI) | $PO_4^{3-}$ | |
| Chitosan | 2.0 | 40 °C | 140 min | 2.48 | - | [37] |
| Chitosan | 4.0 | 30 °C | 40 min | - | 6.65 | [38] |
| Chitosan–zirconia microballs | 5.0 | 30 °C | 60 min | 73.81 | 65.51 | [22] |
| Chitosan–polyacrylamide nanofibers | 6.0 | 45 °C | 60 min | 0.26 | 392 | [39] |
| Chitosan/$Fe_3O_4$/$ZrO_2$ | 3.0 | 25 °C | 240 min | - | 26.5 | [40] |
| Cherry kernel charcoal chitosan composite | 2.0 | 25 °C | 120 min | 14.455 | - | [41] |
| Fe–CTB | 3.0 | 25 °C | 60 min | 34.15 | 32.27 | This study |

### 3.13. Proposed Mechanism of Adsorption

It was clear from the pH studies that the adsorption capacity decreased substantially with an increase in pH for both chromate and phosphate ions. The adsorption efficiencies were more than 80% at pH 3.0, which came down to 40% at pH 7.0. This is a clear indication that the protonation of amine groups of chitosan plays an important role in the adsorption process. Both chromate and phosphate ions behave as anions at pH 4.0, and so there is a strong electrostatic interaction between the protonated amine group of chitosan and these two anions. In short, the forces of adsorption are the strong electrostatic forces between protonated amine groups of the adsorbent and the anionic adsorbate species.

### 4. Conclusions

Fe–CTB showed significant removal with maximum adsorption capacities of 34.15 and 32.27 mg g$^{-1}$ at pH 3.0 for Cr(VI) and phosphate, respectively, which are much higher compared with native chitosan. The Langmuir adsorption isotherm followed in harmony with the pseudo-second-order kinetics for both ions. Additionally, the thermodynamic parameters revealed that the adsorption process of ions by Fe–CTB was exothermic as the value of change in enthalpy was negative. The spontaneity of the adsorption process was explained on the basis of the negative change in Gibbs free energy. The material has been easily synthesized and regenerates easily. Additionally, it is cost valuable and can be reused in multiple cycles for further adsorption. Comparative data against different reported materials prepared from chitosan showed that our newly developed chitosan composite, Fe–CTB, is among the best adsorbents in this category of materials.

**Author Contributions:** S.A.T. and M.A.P.: Methodology and investigation. A.S.: Writing—first draft. E.F.-K. and S.P.: Validation, review, and editing. R.M.J.: Conceptualization and supervision. All authors have read and agreed to the published version of the manuscript.

**Funding:** This research received no external funding.

**Data Availability Statement:** Not applicable.

**Acknowledgments:** The URSP of RTM Nagpur University, bearing no. Dev/2117, is acknowledged. Additionally, thanks, DST, New Delhi, for the DST-FIST grant and UGC, New Delhi, for the UGC-SAP schemes.

**Conflicts of Interest:** The authors declare no conflict of interest.

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
