# Peer review of "Fe(III)–Chitosan Microbeads for Adsorptive Removal of Cr(VI) and Phosphate Ions"

_minerals, doi:10.3390/min12070874_

Round 1

Reviewer 1 Report

This manuscript describing an Fe-chitosan complex used as a sorbent for the removal of Cr(VI) and PO4 covers and important area of pollution control. The most substantial change to be made is to separate the experimental from the R&D section. It is not initially clear how the Cr and PO4 solutions were treated with the Fe-CTB. Another point to be made is at Line 324, the reusability studies. It is not clear that the NaCl actually removed the Cr and PO4 from the Fe-CTB. You need to show that this is the case. It would also be better to characterize the Fe-CTB after the 8th run.

Other changes to consider:

Line 27 drop The in the first line.

Line 42 the Cr(VI) limit is 0.05 mg/l

Line 52 change unpractical to impractical.

Line 57 trials not trails

Line 87 needs a citation

Prior to line 94 you need to ad the experimental describing the adsorption of Cr.

Line 104 change achieved to performed

Lines 119-121 need to be deleted

Line 126 depicted should be assigned.

Could you put the two IR spectra in one frame?

Line 169 and 177 this should be in the experimental section

Line 235 Please give the equations for the isotherms

Line 239 nature rather than natures

Line 281 Please give the equations used here.

Author Response

Author's Reply to the Review Report (Reviewer 1)

Comments and Suggestions for Authors

This manuscript describes a Fe-chitosan complex used as a sorbent for the removal of Cr(VI) and PO4 covers and an important area of pollution control. The most substantial change to be made is to separate the experimental from the R&D section. It is not initially clear how the Cr and PO4 solutions were treated with the Fe-CTB. Another point to be made is at Line 324, the reusability studies. It is not clear that the NaCl actually removed the Cr and PO4 from the Fe-CTB. You need to show that this is the case. It would also be better to characterize the Fe-CTB after the 8th run.

Response: Thank you very much for your valuable suggestions. As per the major change suggested, now we have created a separated section 2.3 in the revised manuscript under the title of Batch Adsorption Experiments. Under this section, the treatment of Cr(VI) and phosphate solutions with Fe-CTB and the working parameter variation has been discussed in details.

Secondly, the action of NaCl on used Fe-CTB was critically studied. The washings obtained from after eighth desorption cycle were subjected to qualitative detection of chromate and phosphate ions. It was observed that the washings contain these ions indicating that there is ion-exchange phenomenon taking place leading to ex-change of chromate or phosphate ions with the chloride ions. This discussion has been incorporated in the revised manuscript.

Other changes to consider:

Line 27 drop The in the first line.

Response: Corrected.

Line 42 the Cr(VI) limit is 0.05 mg/l

Response: Extremely sorry for this mistake. Now it has been rectified.

Line 52 change unpractical to impractical.

Response: Corrected to impractical.

Line 57 trials not trails

Response: Corrected to trials

Line 87 needs a citation

Response: Citation incorporated.

Prior to line 94 you need to add the experimental describing the adsorption of Cr.

Response: This portion is completely modified with incorporation of a new section 2.3 giving all the details of batch adsorption experiments.

Line 104 change achieved to performed

Response: This change has been made in the new section 2.3

Lines 119-121 need to be deleted

Response: Sorry for the sentences remaining from the article template. These lines have been now removed from the revised manuscript.

Line 126 depicted should be assigned.

Response: The word “depicted” has been corrected to “assigned”.

Could you put the two IR spectra in one frame?

Response: These spectra were recorded at different scale (one linear and one non-linear) as can be seen from the x-axis. As we do not have the data points at this time, it was not possible to put them in one frame. Hence, they have been placed in different frames, but in same figure.

Line 169 and 177 this should be in the experimental section

Response: As these two sections include characterization of Fe-CTB and highlight important findings that are a part of results and discussion, they have been placed in section 3. However, the experimental details have been put separately as 2.3 in Experimental section.

Line 235 Please give the equations for the isotherms

Response: Linearized equations of the isotherms are given in the revised manuscript.

Line 239 nature rather than natures

Response: Corrected as nature.

Line 281 Please give the equations used here.

Response: Equations for PFO and PSO models are given in the revised manuscript.

Reviewer 2 Report

Dear in the introduction, the authors lacked to put the most diverse adsorbents that have already been developed in the removal of these adsorbates, since in the literature it is possible to find a wide range. They can cite a few, highlighting the differential of their work.

The parameter adjustments were made linearly. It is already known that the linear estimation of the parameters of the kinetics and isotherms models are negatively affected due to the method, and may not correspond to reality (See the reference). Therefore, it is recommended to authors that the parameters are estimated again, but using the non-linear method.

CASSOL, G.O. ; GALLON, R.; SCHWAAB, M.; BARBOSA-COUTINHO, E. ; SEVERO JR, J.B. ; PINTO, J.C. . Statistical Evaluation of Non-Linear Parameter Estimation Procedures for Adsorption Equilibrium Models. ADSORPTION SCIENCE & TECHNOLOGY, v. 32, p. 257-274, 2014.

I suggest creating a table relating the capacity obtained by the material developed with others present in the literature, according to the model present in these works:

https://doi.org/10.1016/j.cej.2020.125423

https://doi.org/10.1016/j.jece.2020.104574

https://doi.org/10.1016/j.jece.2020.104911

https://doi.org/10.1016/j.jece.2015.11.018

https://doi.org/10.1016/j.powtec.2020.01.064

Finally, the authors must propose a reaction mechanism.

Author Response

Author's Reply to the Review Report (Reviewer 2)

Comments and Suggestions for Authors

  1. Dear in the introduction, the authors lacked to put the most diverse adsorbents that have already been developed in the removal of these adsorbates, since in the literature it is possible to find a wide range. They can cite a few, highlighting the differential of their work.

Response: Variety of adsorbents used in the removal of various dyes have now been cited in the revised manuscript (Please refer references 13-17).

  1. The parameter adjustments were made linearly. It is already known that the linear estimation of the parameters of the kinetics and isotherms models are negatively affected due to the method, and may not correspond to reality (See the reference). Therefore, it is recommended to authors that the parameters are estimated again, but using the non-linear method. CASSOL, G.O. ; GALLON, R.; SCHWAAB, M.; BARBOSA-COUTINHO, E. ; SEVERO JR, J.B. ; PINTO, J.C. . Statistical Evaluation of Non-Linear Parameter Estimation Procedures for Adsorption Equilibrium Models. ADSORPTION SCIENCE & TECHNOLOGY, v. 32, p. 257-274, 2014.

Response: Thank you very much for providing this nice reference. We observed that the linearized Freundlich model fits very well to the obtained data. However, we will definitely try to use non-linear models in our future work.

  1. I suggest creating a table relating the capacity obtained by the material developed with others present in the literature, according to the model present in these works:

https://doi.org/10.1016/j.cej.2020.125423

https://doi.org/10.1016/j.jece.2020.104574

https://doi.org/10.1016/j.jece.2020.104911

https://doi.org/10.1016/j.jece.2015.11.018

https://doi.org/10.1016/j.powtec.2020.01.064

Response: Thank you very much for these excellent references. We have incorporated all of these references in our introduction section. Also, we have modified our Table 4 in accordance with these references for comparing the reported material with other chitosan-based materials for Cr(VI) and phosphate removal.

Finally, the authors must propose a reaction mechanism.

Response: Proposed mechanism has been separately discussed in section 3.13 in the revised manuscript.

Round 2

Reviewer 1 Report

The revised version of the paper is quite good.

Reviewer 2 Report

The manuscript may be accepted for publication.